# DiffCLIP: Differential Attention Meets CLIP

**Hasan Abed Al Kader Hammoud**  *hasanabedalkader.hammoud@kaust.edu.sa*
*King Abdullah University of Science and Technology (KAUST)*

**Bernard Ghanem**  *bernard.ghanem@kaust.edu.sa*
*King Abdullah University of Science and Technology (KAUST)*

**Reviewed on OpenReview:** *https: // openreview. net/ forum? id= 2I2fTehry2*

## Abstract

We propose DiffCLIP, a novel vision-language model that extends the differential attention mechanism to CLIP architectures. Differential attention was originally developed for large language models to amplify relevant context while canceling out noisy information. In this work, we integrate this mechanism into CLIP's dual encoder (image and text) framework. With minimal additional parameters, DiffCLIP achieves superior performance on image-text understanding tasks. Across zero-shot classification, retrieval, and robustness benchmarks, DiffCLIP consistently outperforms baseline CLIP models. Notably, these gains come with negligible computational overhead, demonstrating that differential attention can significantly enhance multi-modal representations without sacrificing efficiency. Code and models can be found at https://github.com/hammoudhasan/DiffCLIP.

## 1 Introduction

Vision-language models (VLMs) have made remarkable progress in bridging the gap between textual and visual modalities, enabling powerful capabilities such as zero-shot image classification, image-text retrieval, and descriptive captioning (Radford et al., 2021; Jia et al., 2021). By aligning images and text in a joint embedding space, these models capture broad semantic relationships across modalities and often excel at out-of-distribution generalization. Among VLMs, Contrastive Language-Image Pre-training (CLIP) (Radford et al., 2021) stands out as a foundational approach, demonstrating strong zero-shot performance on numerous benchmarks with minimal fine-tuning.

While CLIP's contrastive training regime has been widely adopted, its *attention mechanism* can sometimes focus on irrelevant or spurious features in both the image and text encoders. This *attention noise* can hamper fine-grained understanding, particularly when precise localization or explicit contextual knowledge is required. Interestingly, recent language modeling research has proposed a *differential attention* mechanism (Ye et al., 2024), which subtracts complementary attention distributions to suppress noise and highlight salient tokens. However, whether a similar strategy would be effective for multimodal tasks has remained an open question.

> *"Can differential attention be adapted to vision-language models in a way that meaningfully improves their ability to focus on relevant features across modalities?"*

Motivated by this question, we introduce DiffCLIP, an extension of CLIP that integrates differential attention into both the vision and text encoders. By learning two attention maps and subtracting one from the other, DiffCLIP effectively cancels out misaligned or noisy signals, enabling a more precise alignment of images and text. Crucially, this enhancement introduces only a negligible overhead in model parameters and computational cost. Our results show that DiffCLIP consistently outperforms standard CLIP in a wide range of tasks, including linear probing, few-shot classification, image text retrieval, outside-domain robustness, and fine-grained visual understanding, highlighting the efficacy of differential attention in a multimodal

Figure 1: **CC3M Pretraining: CLIP vs. DiffCLIP Across Six Tasks.** We compare standard CLIP (blue) and our DiffCLIP variant (pink) on linear probing, few-shot classification, image/text retrieval, zero-shot ImageNet, and zero-shot OOD. In each case, DiffCLIP consistently outperforms CLIP, highlighting the effectiveness of differential attention with only 0.003% extra parameters.

setting. As shown in Figure 1, DiffCLIP is capable of improving performance across various benchmarks with only 0.003% extra parameters. Figure 2 also shows how DiffCLIP is capable of suppressing attention noise compared to CLIP models with vanilla non-differential attention.

**Our contributions are threefold:**

- We propose DiffCLIP, the first integration of differential attention into CLIP-based VLMs, yielding a simple yet effective approach to reducing attention noise in both vision and text streams.

- Through extensive experiments on Conceptual Captions 3M/12M pretraining, we demonstrate consistent gains over baseline CLIP across a diverse suite of tasks, with a minimal parameter overhead of roughly 0.003%.

- We perform detailed ablations, showing that *(i)* dynamic initialization can boost zero-shot performance, and *(ii)* applying differential attention solely in the vision encoder already captures most of the benefits, suggesting a flexible and cost-effective path to improved multimodal learning.

The remainder of this paper is organized as follows. Section 2 surveys previous work on training-centric, model-centric, and data-centric strategies for enhancing CLIP. Section 3 provides an overview of the standard Transformer attention mechanism, the differential attention concept, and the CLIP framework. Section 4 details our experimental setup, empirical results, and ablation studies, while Sections 5 and 7 concludes with a discussion of future research directions and wrapping up of the paper.

## 2    Related Work

Vision-language pre-training (VLP) has advanced our ability to learn joint representations of images and text, leading to improvements in tasks such as image retrieval, visual question answering, and zero-shot classification (Gan et al., 2022; Zhang et al., 2024). CLIP (Radford et al., 2021) has been central to this progress by using a contrastive loss to align image and text embeddings from large-scale image-caption data.

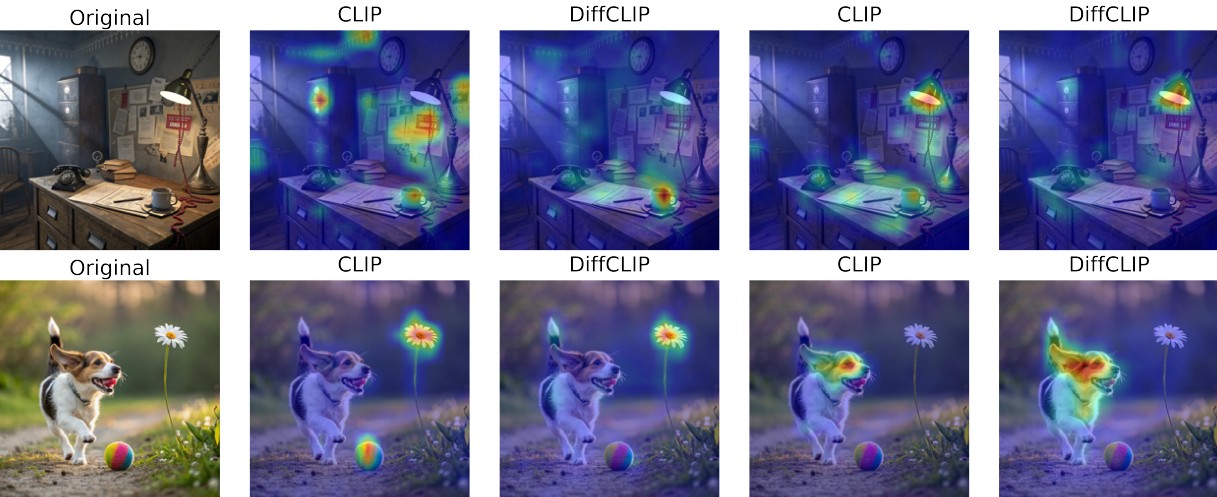

Figure 2: **Comparing CLIP vs. DiffCLIP Attention Maps.** For two images (rows), we visualize where CLIP and DiffCLIP attend when matching each image against two different textual queries. While CLIP allocates attention to irrelevant background regions, DiffCLIP more effectively centers on query-relevant objects, highlighting how differential attention can reduce noise and improve focus.
*Queries: **First Row:** 'Mug", Lamp"; **Second Row:** Flower", Dog".*

Despite CLIP's strong zero-shot performance, researchers continue to explore improvements in its training, architecture, and data collection strategies. These efforts generally fall into three categories: training-centric, model-centric, and data-centric approaches.

**Training-Centric Approaches**  A common strategy is to enrich CLIP's contrastive framework with additional objectives. For example, SLIP (Mu et al., 2022) adds masked image modeling to boost downstream results, while DeCLIP (Li et al., 2021) uses nearest-neighbor supervision to enhance data efficiency. SigLIP (Zhai et al., 2023) replaces the standard softmax temperature with a sigmoid loss, allowing larger batch training and improving generalization and robustness to noisy labels. Retrieval-Enhanced CLIP (Iscen et al., 2023) leverages external memory of image-text pairs at inference, achieving significant gains on fine-grained zero-shot tasks. Further, novel training objectives, such as those proposed by Yang et al. (Yang et al., 2022) and PyramidCLIP (Gao et al., 2022), aggregate information across multiple semantic levels, highlighting the benefit of diversified training signals for improved CLIP performance.

**Model-Centric Approaches**  Another line of work modifies CLIP's architecture for greater efficiency or accuracy. The original CLIP (Radford et al., 2021) employs a Transformer (Vaswani et al., 2017) for text and either a ResNet (He et al., 2016) or Vision Transformer (ViT) (Dosovitskiy et al., 2020) for images. Subsequent studies incorporate ideas from object detection and segmentation to capture finer visual details, such as region-level representations (Xu et al., 2022; Zhong et al., 2022). Recently, ViTamin (Chen et al., 2024) proposed a specialized vision transformer architecture tailored specifically for multimodal models, demonstrating improved zero-shot results compared to standard ViTs under similar training setups. Other researchers attempt to unify image and text encoders into a single Transformer (Tschannen et al., 2022), although this approach is less common. Notably, few methods have altered the core attention mechanism within CLIP. Our work addresses this gap by adapting Differential Attention (Ye et al., 2024), originally proposed for language models, to CLIP's multimodal setting. This adaptation aims to reduce attention noise and enhance representation quality.

**Data-Centric Approaches**  Data-centric methods emphasize improving the size, diversity, and quality of pre-training datasets. Initial efforts focused on scaling datasets (Jia et al., 2021; Radford et al., 2021), while more recent approaches prioritize richer and cleaner supervision. VeCLIP (Lai et al., 2024) uses large language models (LLMs) to generate detailed and enriched captions, enhancing textual supervision.

Similarly, CLIPS (Liu et al., 2024b) utilizes truncated synthetic captions to improve visual grounding and retrieval performance, showing that carefully controlled synthetic textual inputs can surpass standard image-caption pairs. SynthCLIP (Hammoud et al., 2024) explores training entirely on synthetic image-text pairs. Further methods employ filtering techniques to eliminate noisy or irrelevant samples (Gadre et al., 2023; Abbas et al., 2023), while Cluster Masking (Wei et al., 2024) proposes masking clusters of similar image patches, leading to faster training and improved representation quality. These efforts underline the potential of data curation and augmentation strategies in bolstering the efficacy of CLIP-based models.

Beyond performance, fairness and compositionality have also received increased attention. FairCLIP (Luo et al., 2024) addresses demographic biases found in models like CLIP by using optimal-transport-based feature alignment across demographic groups. Meanwhile, iterated learning approaches (Zheng et al., 2024) tackle the compositional limitations of large vision-language models, promoting representations that generalize more reliably to complex and compositional visual-linguistic scenarios.

> In this paper, we contribute to the model-centric direction by adapting Differential Attention (Ye et al., 2024) to CLIP's dual-encoder architecture. Through this adaptation, we aim to reduce attention noise and enhance performance across various image-text understanding tasks.

## 3 Preliminaries

In this section, we outline the fundamental concepts that are essential for our approach. We begin by reviewing the Transformer self-attention mechanism (Vaswani et al., 2017), which is widely used in modern sequence modeling. Next, we introduce differential attention (Ye et al., 2024), a technique designed to reduce attention noise by leveraging complementary attention distributions. Finally, we summarize the Contrastive Language-Image Pre-training (CLIP) framework (Radford et al., 2021), which learns to align images and text in a shared representation space. These components form the basis for our model and experiments.

### 3.1 Transformer Attention

Transformer networks (Vaswani et al., 2017) capture relationships among elements in a sequence through a *self-attention* operation. Let

$$X \in \mathbb{R}^{N \times d}$$

be an input sequence of $N$ tokens (or image patches), each embedded in a $d$-dimensional space. The Transformer maps $X$ to queries ($\mathbf{Q}$), keys ($\mathbf{K}$), and values ($\mathbf{V}$) using learned weight matrices:

$$\mathbf{Q} = X\,W^Q, \quad \mathbf{K} = X\,W^K, \quad \mathbf{V} = X\,W^V,$$

where $W^Q, W^K, W^V \in \mathbb{R}^{d \times d}$. Self-attention scores are then computed via scaled dot-products:

$$A \;=\; \mathrm{softmax}\Big(\tfrac{\mathbf{Q}\mathbf{K}^\top}{\sqrt{d}}\Big),$$

and these scores are used to weight $\mathbf{V}$:

$$\mathrm{Attn}(X) \;=\; A\,\mathbf{V}.$$

To capture different types of relationships, Transformers use *multi-head attention* (MHA). An MHA module with $h$ heads splits each projection into lower-dimensional parts of size $d_h = d/h$. In each head $i$,

$$\mathrm{Attn}_i(X) \;=\; \mathrm{softmax}\Big(\tfrac{\mathbf{Q}_i\mathbf{K}_i^\top}{\sqrt{d_h}}\Big)\mathbf{V}_i,$$

where $\mathbf{Q}_i, \mathbf{K}_i, \mathbf{V}_i \in \mathbb{R}^{N \times d_h}$. The head outputs are concatenated and projected back:

$$\mathrm{MHA}(X) \;=\; \big[\mathrm{Attn}_1(X) \,\|\, \ldots \,\|\, \mathrm{Attn}_h(X)\big]\,W^O,$$

with $W^O \in \mathbb{R}^{(h\,d_h) \times d}$. Despite remarkable success in many areas, standard attention can assign non-negligible weights to irrelevant tokens (often called *attention noise*) (Kamradt, 2023; Liu et al., 2024a), which can degrade performance in settings requiring precise focus.

### 3.2 Differential Attention

Differential attention (Ye et al., 2024) addresses attention noise by learning two separate attention distributions and subtracting one from the other, effectively canceling out spurious alignments.

**Single-Head Differential Attention.** Let $X \in \mathbb{R}^{N \times d}$ be the input to a single attention head. We split **Q** and **K** into two halves, denoted by subscripts 1 and 2:

$$[\mathbf{Q}_1; \mathbf{Q}_2] = X W^Q, \quad [\mathbf{K}_1; \mathbf{K}_2] = X W^K, \quad \mathbf{V} = X W^V,$$

where $\mathbf{Q}_1, \mathbf{Q}_2, \mathbf{K}_1, \mathbf{K}_2 \in \mathbb{R}^{N \times \frac{d}{2}}$. Each half computes its own attention distribution:

$$A_1 = \mathrm{softmax}\Big(\frac{\mathbf{Q}_1 \mathbf{K}_1^\top}{\sqrt{d/2}}\Big), \quad A_2 = \mathrm{softmax}\Big(\frac{\mathbf{Q}_2 \mathbf{K}_2^\top}{\sqrt{d/2}}\Big).$$

The output is formed by subtracting the second distribution (scaled by a learnable parameter $\lambda$) from the first:

$$\mathrm{DiffAttn}(X) = \big(A_1 - \lambda A_2\big) \mathbf{V}.$$

The parameter $\lambda$ is trained to control how strongly the second distribution is subtracted:

$$\lambda = \exp\big(\lambda_{q_1} \lambda_{k_1}\big) - \exp\big(\lambda_{q_2} \lambda_{k_2}\big) + \lambda_{\mathrm{init}},$$

where $\lambda_{q_1}, \lambda_{k_1}, \lambda_{q_2}, \lambda_{k_2}$ are learnable weights and $\lambda_{\mathrm{init}}$ is a hyperparameter. This subtraction often yields a sparser, more focused attention map, which can improve results in scenarios sensitive to background or redundant signals (Ye et al., 2024).

**Multi-Head Extension.** Like standard attention, differential attention can be extended to multiple heads. In *Differential Multi-Head Attention* (Diff MHA), each head $i$ applies the differential step independently:

$$\mathrm{DiffAttn}_i(X) = \left(\mathrm{softmax}\Big(\frac{\mathbf{Q}_{1,i} \mathbf{K}_{1,i}^\top}{\sqrt{d_h/2}}\Big) - \lambda \, \mathrm{softmax}\Big(\frac{\mathbf{Q}_{2,i} \mathbf{K}_{2,i}^\top}{\sqrt{d_h/2}}\Big)\right) \mathbf{V}_i,$$

where $\mathbf{Q}_{1,i}, \mathbf{Q}_{2,i}, \mathbf{K}_{1,i}, \mathbf{K}_{2,i} \in \mathbb{R}^{N \times (d_h/2)}$. The final output is then

$$\mathrm{DiffMHA}(X) = \big[\mathrm{DiffAttn}_1(X) \,\|\, \dots \,\|\, \mathrm{DiffAttn}_h(X)\big] W^O.$$

By learning complementary attention maps in each head and subtracting them, Diff MHA aims to amplify relevant patterns while reducing noise.

### 3.3 CLIP Training

Contrastive Language-Image Pre-training (CLIP) (Radford et al., 2021) learns image and text embeddings in a shared space using a large collection of paired image-text examples $\{(I_k, T_k)\}_{k=1}^M$. It consists of two encoders: one for images $(f_\theta)$ and one for text $(g_\phi)$. Their outputs are normalized to unit length:

$$u_i = \frac{f_\theta(I_i)}{\|f_\theta(I_i)\|_2}, \quad v_i = \frac{g_\phi(T_i)}{\|g_\phi(T_i)\|_2}.$$

For a batch of $N$ pairs, CLIP forms a similarity matrix

$$S_{ij} = \frac{u_i^\top v_j}{\tau},$$

where $\tau$ is a (learned or fixed) temperature parameter. The text-to-image contrastive loss is

$$\mathcal{L}_{ti} = -\frac{1}{N} \sum_{i=1}^N \log \frac{\exp(S_{ii})}{\sum_{j=1}^N \exp(S_{ij})},$$

Table 1: **Classification Performance (Linear Probing and Few-Shot).** We compare CLIP and DiffCLIP on nine classification tasks with two pretraining sets (CC3M and CC12M). The top block reports linear probing accuracy, while the bottom block shows few-shot results. Numbers in parentheses indicate absolute gains or drops for DiffCLIP relative to CLIP.

| Pretraining | Model | Caltech-101 | DTD | Pets | Flowers | SUN397 | Aircraft | CIFAR10 | CIFAR100 | Food-101 | Avg. |
|---|---|---|---|---|---|---|---|---|---|---|---|
| **Linear Probing** | | | | | | | | | | | |
| CC3M | CLIP | 72.5 | 58.7 | 61.0 | 85.8 | 54.1 | 35.7 | 83.5 | 63.4 | 59.1 | 63.8 |
| CC3M | DiffCLIP | 76.2 (+3.7) | 60.2 (+1.5) | 62.2 (+1.2) | 86.6 (+0.8) | 56.2 (+2.1) | 34.6 (-1.1) | 83.9 (+0.4) | 63.7 (+0.3) | 59.4 (+0.3) | 64.8 (+1.0) |
| CC12M | CLIP | 88.3 | 71.2 | 79.5 | 92.6 | 68.3 | 48.8 | 92.0 | 74.7 | 77.5 | 77.0 |
| CC12M | DiffCLIP | 89.5 (+1.2) | 71.8 (+0.6) | 83.0 (+3.5) | 93.5 (+0.9) | 69.4 (+1.1) | 46.4 (-2.4) | 90.7 (-1.3) | 73.3 (-1.4) | 77.7 (+0.2) | 77.3 (+0.3) |
| **Few-Shot** | | | | | | | | | | | |
| CC3M | CLIP | 90.4 | 72.9 | 69.6 | 92.5 | 91.8 | 44.6 | 63.4 | 72.8 | 67.0 | 73.9 |
| CC3M | DiffCLIP | 91.6 (+1.2) | 73.2 (+0.3) | 71.6 (+2.0) | 92.9 (+0.4) | 92.8 (+1.0) | 45.4 (+0.8) | 62.4 (-1.0) | 73.5 (+0.7) | 68.3 (+1.3) | 74.6 (+0.7) |
| CC12M | CLIP | 97.4 | 81.9 | 86.3 | 96.9 | 96.5 | 56.1 | 81.3 | 85.1 | 86.0 | 85.3 |
| CC12M | DiffCLIP | 97.6 (+0.2) | 82.2 (+0.3) | 88.2 (+1.9) | 97.3 (+0.4) | 96.8 (+0.3) | 55.2 (-0.9) | 80.3 (-1.0) | 83.3 (-1.8) | 87.5 (+1.5) | 85.4 (+0.1) |

Table 2: **Zero-Shot Retrieval and ImageNet Zero-shot Accuracy.** We report image and text retrieval (Recall@5, %) and zero-shot ImageNet accuracy (%) for CLIP vs. DiffCLIP, using CC3M or CC12M as pretraining data. Values in parentheses reflect absolute gains or drops for DiffCLIP relative to CLIP.

| Pretraining | Model | Image Retrieval (R@5) | | | | Text Retrieval (R@5) | | | | Zero-Shot |
|---|---|---|---|---|---|---|---|---|---|---|
| | | Flickr30k | Flickr8k | MSCOCO | Avg. | Flickr30k | Flickr8k | MSCOCO | Avg. | ImageNet |
| CC3M | CLIP | 31.8 | 35.4 | 19.4 | 28.9 | 43.4 | 46.2 | 25.4 | 38.3 | 13.6 |
| CC3M | DiffCLIP | 32.9 (+1.1) | 36.5 (+1.1) | 20.9 (+1.5) | 30.1 (+1.2) | 44.7 (+1.3) | 47.8 (+1.6) | 27.6 (+2.2) | 40.1 (+1.8) | 14.4 (+0.8) |
| CC12M | CLIP | 62.5 | 62.1 | 41.3 | 55.3 | 76.8 | 77.7 | 53.8 | 69.4 | 31.8 |
| CC12M | DiffCLIP | 62.2 (-0.3) | 61.5 (-0.6) | 42.3 (+1.0) | 55.3 (+0.0) | 77.4 (+0.6) | 77.4 (-0.3) | 55.5 (+1.7) | 70.1 (+0.7) | 33.8 (+2.0) |

and the image-to-text counterpart is

$$\mathcal{L}_{it} = -\frac{1}{N} \sum_{i=1}^{N} \log \frac{\exp(S_{ii})}{\sum_{j=1}^{N} \exp(S_{ji})}.$$

The overall objective is

$$\mathcal{L}_{\text{CLIP}} = \frac{1}{2}\Big(\mathcal{L}_{ti} + \mathcal{L}_{it}\Big).$$

By encouraging matching image-text pairs to have high similarity (and non-matching pairs to have low similarity), CLIP learns robust features that often transfer well to downstream tasks like zero-shot classification and retrieval.

# 4 Experiments

We present an extensive empirical study to investigate whether differential attention can benefit CLIP-style vision-language models. We first describe our dataset sources and training configurations, then evaluate both standard CLIP and our DiffCLIP variant under linear probing, few-shot classification, and image-text retrieval. We also test robustness to distribution shifts (via OOD ImageNet) and fine-grained features (via MMVP), and conclude with ablation studies on the initialization of the differential attention parameter $\lambda_{\text{init}}$ and on applying differential attention to only the vision encoder.

## 4.1 Experimental Setup

**Datasets.** We pretrain on Conceptual Captions 3M (CC3M) (Sharma et al., 2018) and Conceptual Captions 12M (CC12M) (Changpinyo et al., 2021). After downloading using `img2dataset` (Beaumont,

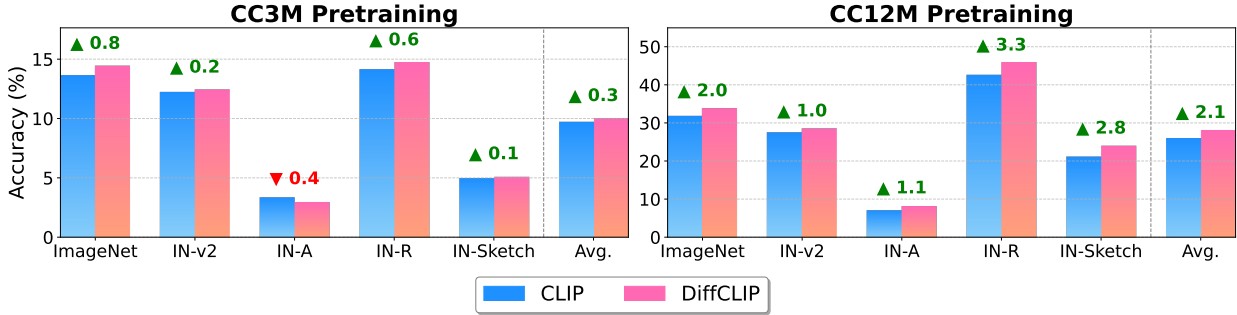

Figure 3: **OOD Zero-Shot ImageNet Performance.** Comparison of zero-shot accuracy (%) on ImageNet, ImageNet-V2, ImageNet-A, ImageNet-R, and ImageNet-Sketch, plus the average. Bars show performance of CLIP (blue) versus DiffCLIP (pink), trained on CC3M (left) or CC12M (right). Numerical deltas above the bars indicate the absolute improvement or drop for DiffCLIP relative to CLIP. DiffCLIP improves on average the zero-shot performance on OOD ImageNet datasets as compared to CLIP.

2021) (with shorter edge resized to 224), we end up with about 2.3M image-text pairs for CC3M and 7.9M for CC12M. For CC3M, we train on four A100 GPUs, while CC12M uses eight A100 GPUs to reduce training time. Text data is minimally processed, limited to basic tokenization.

**Training Parameters.** All models train for 40 epochs, using one epoch of linear warmup, a global batch size of 4096, and AdamW optimizer (Loshchilov & Hutter, 2017). We set the base learning rate to $5 \times 10^{-4}$ with weight decay of 0.5. For DiffCLIP, every attention layer in both the vision and text encoders is replaced with differential attention. We initialize each layer's $\lambda$ at 0.8 unless stated otherwise. This setup introduces only a minor parameter overhead: roughly 0.003% additional parameters relative to a standard CLIP-B/16. Training parameters are chosen similar to SynthCLIP (Hammoud et al., 2024) and training code is adopted from SLIP (Mu et al., 2022).

**Evaluation Protocol.** We follow established practices for linear probing and few-shot evaluation (El Banani et al., 2023) on nine image-classification datasets: DTD (Cimpoi et al., 2014), Flowers (Nilsback & Zisserman, 2008), Pets, Caltech-101 (Li et al., 2022), Aircraft (Maji et al., 2013), CIFAR-10 (Krizhevsky et al., 2009), SUN397 (Xiao et al., 2010), CIFAR-100 (Krizhevsky et al., 2009), and Food-101 (Bossard et al., 2014). For retrieval (image-to-text and text-to-image) on Flickr8k (Rashtchian et al., 2010), Flickr30k (Young et al., 2014), and MSCOCO (Lin et al., 2014), we use the LAION CLIP Benchmark framework (Schuhmann et al., 2022). We measure zero-shot robustness on ImageNet (Russakovsky et al., 2015) and its variants (ImageNet-V2 (Recht et al., 2019), ImageNet-A (Hendrycks et al., 2021b), ImageNet-R (Hendrycks et al., 2021a), and ImageNet-Sketch (Wang et al., 2019)). Finally, we use the MMVP-VLM benchmark (Tong et al., 2024) to check how well each model focuses on fine-grained visual details.

## 4.2 Do CLIP Models Benefit from Differential Attention?

**Motivation.** To evaluate the effectiveness of our proposed DiffCLIP, we test its performance across tasks involving image classification, image-text retrieval, and zero-shot generalization, following common benchmarks established in prior literature (Hammoud et al., 2024).

**Results.** We compare a baseline CLIP-B/16 to our DiffCLIP-B/16 (with differential attention in both vision and text encoders). Table 1 shows linear probing and few-shot classification results for models pretrained on both CC3M and CC12M. DiffCLIP outperforms standard CLIP on almost every dataset. For example, with CC3M pretraining, DiffCLIP achieves about +1% gain in linear probing and +0.7% in few-shot accuracy.

Table 2 presents retrieval metrics and zero-shot ImageNet. DiffCLIP again surpasses CLIP on image and text retrieval: for CC3M, we see an average improvement of about 1.2% (image retrieval) and 1.8% (text retrieval). On zero-shot ImageNet, DiffCLIP-CC3M increases accuracy by 0.8%, with even larger gains of +2.0% when using CC12M.

> **Conclusion.** Even though DiffCLIP only adds a tiny fraction of extra parameters, it consistently outperforms standard CLIP on classification and retrieval benchmarks. This suggests that differential attention is a lightweight yet effective way to enhance vision-language representation.

### 4.3 Does Differential Attention Improve Out-of-Domain Robustness?

**Motivation.** Having observed improvements on in-distribution ImageNet, we ask if these gains carry over to more challenging out-of-domain variants. Real-world applications often involve domain shifts, and CLIP's zero-shot adaptability has been tested on ImageNet-V2, ImageNet-A, ImageNet-R, and ImageNet-Sketch—benchmarks known to stress model robustness beyond standard ImageNet. Understanding how differential attention influences robustness in such scenarios is crucial for assessing its practical utility in deployment settings. We aim to see if differential attention helps maintain or improve performance under such shifts.

**Results.** Figure 3 summarizes zero-shot performance across ImageNet-V2, ImageNet-A, ImageNet-R, and ImageNet-Sketch. Models with differential attention outperform standard CLIP by an average of 2.1%, suggesting that subtracting noisy attention patterns yields features that generalize more robustly, even under significant distribution shifts.

> **Conclusion.** DiffCLIP not only enhances in-distribution performance but also strengthens zero-shot robustness against substantial domain shifts, further demonstrating the benefits of differential attention.

### 4.4 Does DiffCLIP Improve Fine-Grained Visual Understanding?

**MMVP-VLM Benchmark.** To test fine-grained visual understanding, we employ the MMVP-VLM benchmark (Tong et al., 2024). This benchmark measures how well vision-language models capture nuanced visual properties, such as object orientation, presence, and relational context, beyond straightforward recognition. Both CLIP and DiffCLIP are pretrained on CC12M under identical settings.

**Results.** On average, DiffCLIP improves MMVP-VLM accuracy by 5.7% relative to baseline CLIP. A radar plot (Figure 4) shows DiffCLIP surpassing or matching CLIP on nearly all categories except one (*state and condition*). This suggests that subtracting noisy attention patterns (via differential attention) helps the model attend to more subtle details in images.

**Interpretation.** The MMVP benchmark is specifically designed to require precise visual focus on relevant object details in order to answer fine-grained questions correctly. Therefore, the observed +5.7% absolute improvement over baseline CLIP provides strong quantitative evidence that DiffCLIP's reduced attentional noise translates directly into more accurate and discriminative visual focus. This aligns with our qualitative visualizations, where DiffCLIP consistently attends to the most semantically relevant regions, supporting our central claim that differential attention enhances the quality of attention maps in CLIP-style models.

> **Conclusion.** By mitigating extraneous context through differential attention, DiffCLIP achieves stronger fine-grained visual understanding. These gains highlight the effectiveness of explicitly canceling irrelevant attention weights in multimodal settings.

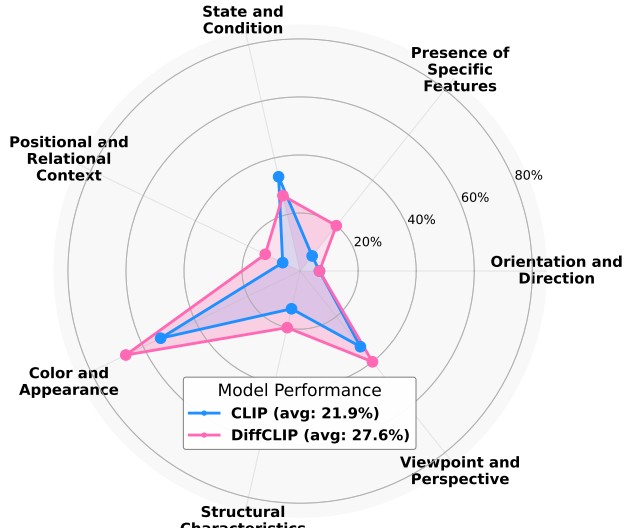

Figure 4: **MMVP-VLM Benchmarking.** Radar plot illustrating performance on different fine-grained visual categories. Both models (CLIP in blue, DiffCLIP in pink) are evaluated on properties like orientation, positional context, and color appearance. DiffCLIP (average 27.6%) consistently outperforms CLIP (average 21.9%), demonstrating more focused attention on subtle visual details.

### 4.5 Dynamic or Static $\lambda_{\text{init}}$?

**Motivation.** All previous experiments used a fixed initialization $\lambda_{\text{init}} = 0.8$ for differential attention. However, (Ye et al., 2024) proposes a *dynamic* schedule:

$$\lambda_{\text{init}}(l) = 0.8 - 0.6 \exp(-0.3\,l),$$

where $l$ is the layer index. We denote the model using this schedule as DiffCLIP$^*$.

**Results.** Figure 5 summarizes six tasks: linear probing, few-shot classification, image retrieval, text retrieval, zero-shot ImageNet, and zero-shot OOD. Compared to the baseline CC12M CLIP, DiffCLIP$^*$ improves zero-shot ImageNet by +2.8% and text retrieval by +1.5%. It also raises zero-shot OOD accuracy by +1.3%. However, relative to *standard* DiffCLIP (with fixed $\lambda_{\text{init}} = 0.8$), DiffCLIP$^*$ is +0.8% better on zero-shot ImageNet and +0.8% on text retrieval, but it *underperforms or only slightly improves* on other tasks. For instance, in zero-shot OOD, DiffCLIP$^*$ is -0.8% behind standard DiffCLIP.

> **Conclusion.** A dynamic $\lambda$ schedule yields notable gains on zero-shot ImageNet and text retrieval, though it lags behind the simpler constant initialization on several other benchmarks. Future work might explore how best to tune or combine these schedules to achieve consistent improvements.

### 4.6 Does Applying Differential Attention to Vision Only Suffice?

**Motivation.** Because the vision encoder often plays a dominant role in CLIP models, one might ask whether differential attention is *necessary* in both encoders. We define DiffCLIP$^\dagger$ as a variant that integrates differential attention *only in the vision* encoder, leaving the text encoder with regular attention.

**Results.** Figure 5 compares CLIP, DiffCLIP, and DiffCLIP$^\dagger$ across six tasks: linear probing, few-shot classification, image retrieval, text retrieval, zero-shot ImageNet, and zero-shot OOD. DiffCLIP$^\dagger$ improves upon baseline CLIP by +0.1% in linear probing, +0.3% in few-shot, +0.4% in image retrieval, +1.2% in text

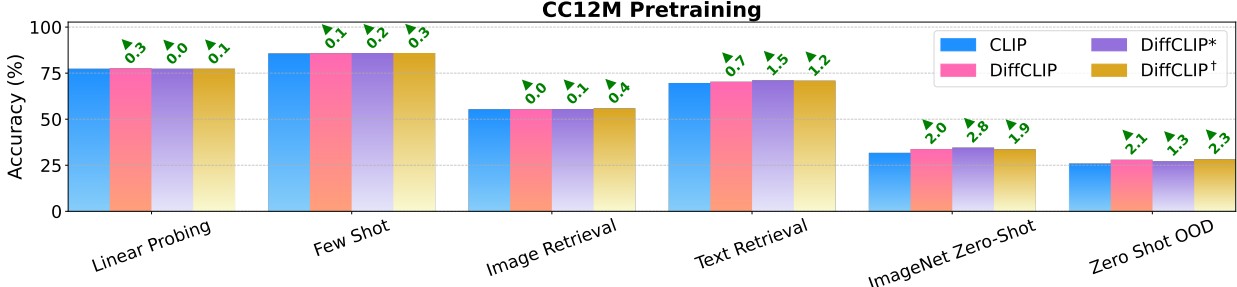

Figure 5: **Comparing Different DiffCLIP Variants.** We evaluate four models on six tasks (linear probing, few-shot, image retrieval, text retrieval, ImageNet zero-shot, and zero-shot OOD), all pretrained on CC12M. CLIP (blue) is the baseline, DiffCLIP (pink) uses a fixed differential attention parameter, DiffCLIP* (purple) employs a dynamic schedule for differential attention, and DiffCLIP† (yellow) applies differential attention only to the vision encoder.

retrieval, +1.9% on zero-shot ImageNet, and +2.3% on zero-shot OOD. Compared to DiffCLIP, DiffCLIP† surpasses or matches performance on few-shot, image retrieval, text retrieval, and zero-shot OOD, but is slightly behind on linear probing and standard zero-shot ImageNet.

> **Conclusion.** Applying differential attention solely to the vision encoder already brings sizable gains. Interestingly, DiffCLIP† can even match or exceed full DiffCLIP on certain tasks, suggesting that most of the performance boost may come from more robust visual feature extraction.

## 5 Future Directions & Limitations

### 5.1 Beyond CLIP

An intriguing question for future research is how a vision encoder trained with differential attention within the CLIP framework would perform when integrated into larger, more sophisticated vision-language models such as LLaVA (Liu et al., 2023) or TinyLLaVA (Zhou et al., 2024). To provide initial insights into this possibility, we conducted preliminary experiments by combining our DiffCLIP-CC12M vision encoder with the Qwen-2.5-Instruct-0.5B (Yang et al., 2024) language encoder. We followed a typical two-stage training procedure: first, a linear projector was trained to align visual tokens with the language embedding space, freezing all other components; second, both the projector and the language encoder underwent instruction fine-tuning.

For the projection training, we utilized the LAION-CC-SBU dataset (558K image-text pairs) used in the LLaVA training setup. For instruction fine-tuning, we adopted the COCO (Lin et al., 2014) subset (approximately 350K pairs) also used by LLaVA. All experiments were conducted using the TinyLLaVA repository

Table 3: **CLIP vs. DiffCLIP on POPE Hallucination Benchmark.** We compare models across three POPE categories, showing accuracy, precision, and recall. Absolute improvements of DiffCLIP over CLIP are highlighted in parentheses.

| | CLIP | | | DiffCLIP | | |
|---|---|---|---|---|---|---|
| **POPE** | Accuracy | Precision | Recall | Accuracy | Precision | Recall |
| **Random** | 50.14 | 50.07 | 98.21 | 50.41 (+0.27) | 50.21 (+0.14) | 98.56 (+0.35) |
| **Popular** | 50.27 | 50.13 | 99.33 | 50.27 (+0.00) | 50.13 (+0.00) | 99.47 (+0.14) |
| **Adversarial** | 50.17 | 50.08 | 99.33 | 50.20 (+0.03) | 50.10 (+0.02) | 99.47 (+0.14) |

on 4 A100-80GB GPUs. The hyperparameters for fine-tuning included a batch size of 48 samples per GPU, a learning rate of $2 \times 10^{-5}$, zero weight decay, a warm-up ratio of 0.03, and cosine decay scheduling. The projection pretraining similarly employed 48 samples per GPU, a learning rate of $1 \times 10^{-3}$, no weight decay, a warm-up ratio of 0.03, and cosine decay scheduling.

We evaluated the resulting models on the POPE (Li et al., 2023) hallucination dataset, which assesses models' susceptibility to visual hallucinations. Despite the modest size of observed improvements, DiffCLIP-CC12M consistently outperformed the CLIP-CC12M baseline across all metrics (Table 3). These initial findings suggest that differential attention-trained vision encoders could enhance performance when integrated into broader vision-language architectures, making it a promising direction for further exploration.

### 5.2 Scaling Data and Architecture

Training CLIP models with a ViT-B/16 backbone on the CC12M dataset (7.9M samples) currently requires approximately 10 GPU-days on A100 GPUs, translating to roughly \$600 using Google Cloud Platform (GCP). A natural future direction would involve exploring how differential attention performs when scaling to larger architectures (e.g., ViT-L or ViT-H) and substantially bigger datasets (e.g., LAION-400M). Investigating such scaling could reveal whether the performance gains observed with DiffCLIP persist or even amplify as model size and dataset scale increase, offering insights into the broader applicability and benefits of differential attention in vision-language pretraining.

## 6 Limitations and Future Work

Although DiffCLIP demonstrates consistent gains in a wide range of tasks, several limitations remain. First, due to computational constraints inherent to academic settings, we are unable to perform large-scale multi-seed pre-training runs (e.g., on LAION-400M). Instead, we focus on smaller scale datasets and we provide statistical measures only for nondeterministic evaluations (linear probing, few-shot), and we rely on the breadth and consistency of results as supporting evidence. Second, while preliminary results on integrating DiffCLIP-trained encoders into larger vision-language models (e.g., LLaVA-style) are promising, exploring this direction at scale remains a future work. We believe that these extensions, particularly scaling to larger datasets and architectures, will further reveal the potential of differential attention in multimodal learning.

## 7 Conclusion

We introduced DiffCLIP, which integrates differential attention into CLIP-based vision-language models to better filter out noisy alignments. Through extensive experiments on classification, retrieval, robustness, and fine-grained benchmarks, DiffCLIP consistently improves over standard CLIP with minimal overhead. Further ablations highlight the flexibility of dynamic attention schedules and vision-only setups. We hope these findings inspire future research on more efficient, robust attention mechanisms in large multimodal learning.

## 8 Acknowledgements

The research reported in this publication was supported by funding from King Abdullah University of Science and Technology (KAUST) - Center of Excellence for Generative AI, under award number 5940.

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

## A    Extra Visualizations

In this appendix, we provide more visualizations of the attention maps similar to Figure 2.

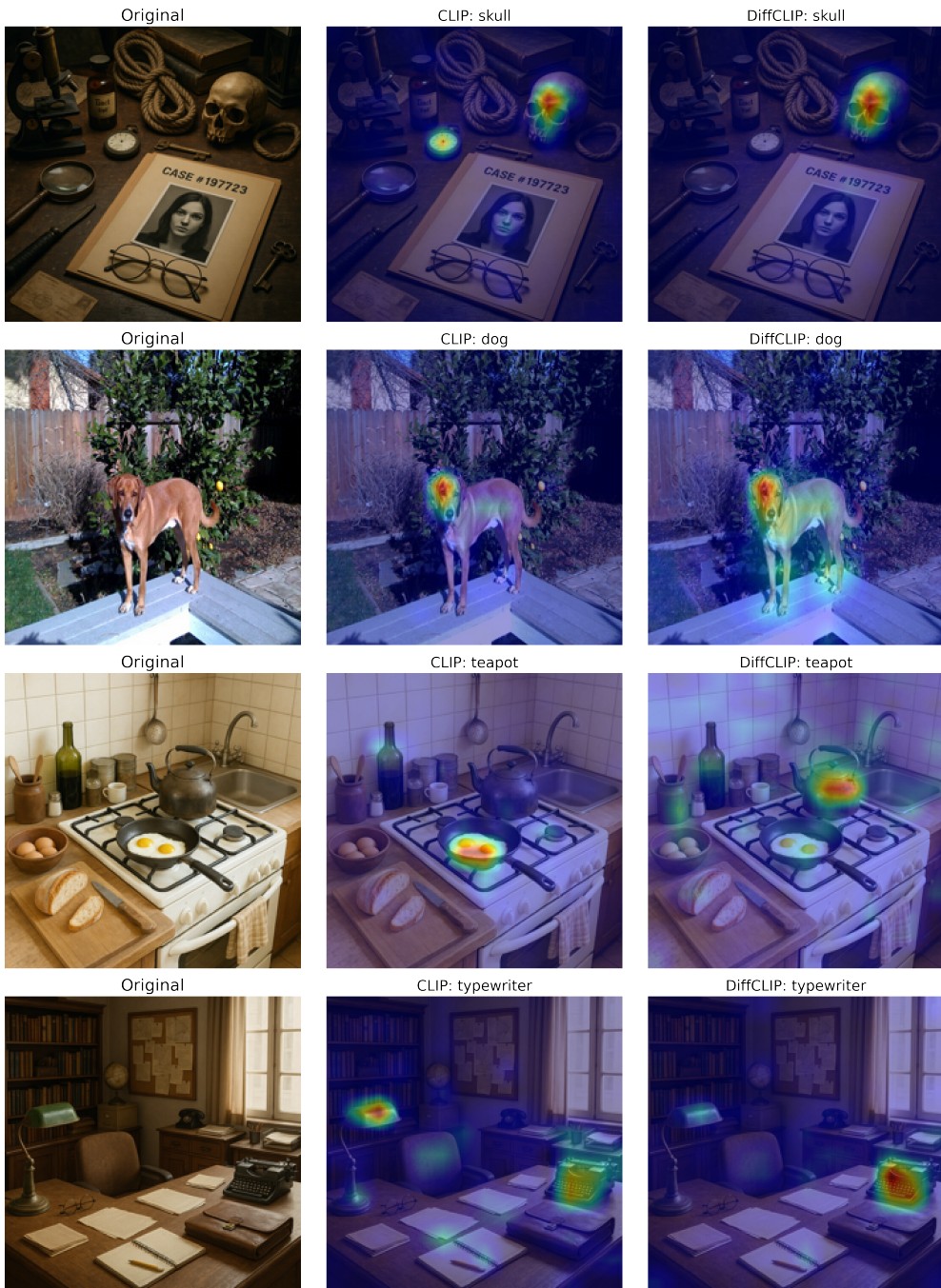

Figure 6: Additional Visualizations of DiffCLIP Attention vs CLIP Attention

**On the Difficulty of Quantitatively Evaluating Attention Maps.**    While qualitative comparisons (as in Figure 6) clearly show that DiffCLIP produces more focused and object-centric attention patterns than CLIP, turning these visual differences into a robust quantitative metric is challenging. A common approach would be to compare attention maps to ground-truth object masks (e.g., from segmentation datasets) using

IoU scores. However, this requires converting soft attention maps into binary masks, which introduces several issues: (i) selecting a single threshold is problematic because optimal values vary widely across images, especially between small and large objects; (ii) attention scores may be diffuse by design for some heads, making them poorly suited to binary evaluation; and (iii) per-image threshold tuning risks introducing bias and undermining reproducibility. Given these constraints, we rely instead on downstream task performance (e.g., the +5.7% gain on the MMVP benchmark) as an indirect but more meaningful quantitative indicator that the attention mechanism improves focus on semantically relevant regions.

## B   Statistical Significance

In this appendix, we run the non-deterministic linear probing and few-shot evaluation from Table 1 on three seeds and report the mean and standard deviation performance.

The following tables (4 and 5) present the aggregated results, enabling a clearer comparison that accounts for non-determinism.

Table 4: **Three-seed Linear Probing (mean $\pm$ sd, %).** Each cell reports the average accuracy and standard deviation over three seeds for non-deterministic evaluations.

| Pretraining | Model | Caltech-101 | DTD | Pets | Flowers | SUN397 | Aircraft | CIFAR10 | CIFAR100 | Food-101 | Avg. |
|---|---|---|---|---|---|---|---|---|---|---|---|
| CC3M | CLIP | $72.5 \pm 0.1$ | $58.7 \pm 0.0$ | $61.0 \pm 0.0$ | $85.8 \pm 0.0$ | $54.1 \pm 0.0$ | $35.6 \pm 0.0$ | $83.5 \pm 0.0$ | $63.4 \pm 0.0$ | $59.1 \pm 0.0$ | $63.7 \pm 0.0$ |
| CC3M | DiffCLIP | $76.2 \pm 0.0$ | $60.2 \pm 0.0$ | $62.2 \pm 0.0$ | $86.6 \pm 0.0$ | $56.2 \pm 0.0$ | $34.6 \pm 0.0$ | $83.9 \pm 0.0$ | $63.7 \pm 0.0$ | $59.4 \pm 0.0$ | $64.8 \pm 0.0$ |
| CC12M | CLIP | $88.4 \pm 0.2$ | $71.2 \pm 0.0$ | $79.9 \pm 0.8$ | $92.6 \pm 0.1$ | $68.3 \pm 0.0$ | $48.8 \pm 0.0$ | $92.0 \pm 0.0$ | $74.7 \pm 0.0$ | $77.5 \pm 0.0$ | $77.1 \pm 0.1$ |
| CC12M | DiffCLIP | $89.5 \pm 0.0$ | $71.8 \pm 0.0$ | $83.0 \pm 0.0$ | $93.5 \pm 0.0$ | $69.4 \pm 0.0$ | $46.8 \pm 0.0$ | $90.7 \pm 0.0$ | $73.3 \pm 0.0$ | $77.7 \pm 0.0$ | $77.3 \pm 0.0$ |

Table 5: **Three-seed Few-Shot (mean $\pm$ sd, %).** Average accuracy and standard deviation over three seeds.

| Pretraining | Model | Caltech-101 | DTD | Pets | Flowers | SUN397 | Aircraft | CIFAR10 | CIFAR100 | Food-101 | Avg. |
|---|---|---|---|---|---|---|---|---|---|---|---|
| CC3M | CLIP | $90.3 \pm 0.4$ | $72.5 \pm 0.5$ | $70.6 \pm 0.2$ | $92.6 \pm 0.4$ | $91.7 \pm 0.1$ | $44.6 \pm 0.3$ | $63.6 \pm 0.2$ | $72.6 \pm 0.3$ | $66.9 \pm 0.5$ | $73.9 \pm 0.1$ |
| CC3M | DiffCLIP | $91.4 \pm 0.3$ | $72.9 \pm 0.5$ | $72.6 \pm 0.2$ | $93.0 \pm 0.3$ | $92.7 \pm 0.2$ | $45.3 \pm 0.3$ | $62.6 \pm 0.2$ | $73.2 \pm 0.4$ | $68.3 \pm 0.3$ | $74.7 \pm 0.1$ |
| CC12M | CLIP | $97.3 \pm 0.1$ | $81.9 \pm 0.4$ | $86.8 \pm 0.2$ | $96.5 \pm 0.3$ | $96.5 \pm 0.2$ | $56.5 \pm 0.3$ | $80.7 \pm 0.2$ | $84.8 \pm 0.2$ | $86.1 \pm 0.1$ | $85.2 \pm 0.0$ |
| CC12M | DiffCLIP | $97.6 \pm 0.1$ | $82.2 \pm 0.4$ | $88.6 \pm 0.3$ | $97.1 \pm 0.2$ | $96.6 \pm 0.1$ | $55.1 \pm 0.2$ | $79.8 \pm 0.1$ | $83.2 \pm 0.2$ | $87.4 \pm 0.1$ | $85.3 \pm 0.0$ |

## C   Intuitive Explanation of Differential Attention

While Section 3.2 presents the formal definition of differential attention, here we offer a more intuitive explanation to complement the mathematical formulation.

Differential attention learns two separate attention maps, $A_1$ and $A_2$, for the same input:

$$A_1 = \text{softmax}\left(\frac{\mathbf{Q}_1 \mathbf{K}_1^\top}{\sqrt{d/2}}\right), \tag{1}$$

$$A_2 = \text{softmax}\left(\frac{\mathbf{Q}_2 \mathbf{K}_2^\top}{\sqrt{d/2}}\right). \tag{2}$$

The final attention output is computed as:

$$\text{DiffAttn}(X) = (A_1 - \lambda A_2)\,\mathbf{V}. \tag{3}$$

**Interpretation.**

- $A_1$ is encouraged by the contrastive objective to highlight the most salient, task-relevant features (e.g., the "dog" in an image).

- $A_2$ is learned through separate projections and, via the subtraction, is implicitly encouraged to capture more diffuse, high-entropy patterns—often corresponding to *noise* such as background textures ("grass," "sky") or features spuriously correlated with the main object.

- Subtracting a scaled version of $A_2$ from $A_1$ reduces these irrelevant activations, producing a sparser and more discriminative attention distribution.

This mechanism helps the model *cancel out* noisy or distracting visual patterns while amplifying the signals that are most relevant to the text query. In practice, we find that this results in attention maps that more tightly focus on semantically important regions, which contributes to the improved performance and robustness observed in Section 4.

