# OpenReview forum: "DiffCLIP: Differential Attention Meets CLIP"
_TMLR — Accepted by TMLR_

### Review · Reviewer_mLBQ · 2025-06-03

**Summary Of Contributions:**

This paper proposes DiffCLIP, a CLIP model using the differential attention mechanism, which was recently proposed in the LLM literature.
The experimental results show that DiffCLIP improves performance on various tasks compared to the counterpart CLIP.

**Audience:**

Yes

**Claims And Evidence:**

Yes

**Requested Changes:**

## Major
- For W1 and W3: Demonstrating that the vanilla CLIP has poor localization, such as in Fig. 2, would support the claim. It would also be interesting to add more examples of failure cases of attention and quantitative evaluation of localization and spurious features and how DiffCLIP alleviates them.
- For W2: A possible explanation or analysis for why DiffCLIP underperformed the vanilla CLIP on Aircraft and CIFAR10/100 in Tab. 1 would provide deeper insights and help clarify the limitations.
- For W3: Providing experimental details and discussions for Fig. 2 and Tab. 3 would be beneficial.
- For W4: Evaluating and visualizing the modality gap in the embedding space would provide more insights for CLIP.


## Minor
- Bar plots such as Figs. 1, 3, and 5 should be replaced with tables. The score differences are generally small (~1 percentage point), making the bar lengths visually indistinguishable. Presenting the exact numerical values would be more informative.
- Numbering all equations would facilitate easier referencing and improve the readability of the technical discussion.

**Strengths And Weaknesses:**

## Strengths
Attention noise and spurious features are critical problems in vision-language tasks.
Showing that the differential attention mitigates attention noise in CLIP is also beneficial for practitioners.

## Weaknesses
- W1: The Introduction claims that attention noise in CLIP leads to poor performance on localization and fine-grained tasks. However, this statement lacks supporting evidence, such as citations or preliminary experiments, to substantiate the claim.
- W2: The experimental evaluation is rather superficial; it merely mentions an improvement in average accuracy without providing a deeper analysis or insights.
- W3: Fig. 2 and Tab. 3 are not referenced or discussed in the experimental section, leaving the reader unclear about their relevance or how they support the paper's findings.
- W4: An analysis of CLIP-specific perspectives, such as modality gap[a,b,c], which is known to be critical for CLIP's generalization, would be intriguing.


[a] Liang et al., Mind the Gap: Understanding the Modality Gap in Multi-modal Contrastive Representation Learning, NeurIPS 2022.
[b] Yamaguchi et al., Post-pre-training for Modality Alignment in Vision-Language Foundation Models, CVPR 2025.
[c] Eslami, et al., Mitigate the Gap: Improving Cross-Modal Alignment in CLIP, ICLR 2025.

---

> ### Author Response · Authors · 2025-07-03
>
> We thank the reviewer for their time and feedback. We will address each of the points raised.
>
> ---
>
> 1- **Lack of supporting evidence for the claim of attention noise.**
>
> We must respectfully but firmly disagree with this assessment. The claim that attention mechanisms can focus on irrelevant or spurious features is a well-established concept in the literature, which we cite (e.g., [A,B] and another that we didn't cite [C]). More importantly, our paper provides extensive evidence to substantiate this claim specifically for CLIP, both qualitatively and quantitatively.
>
> - **Qualitative Evidence:** Figure 2, which the reviewer later notes is unreferenced (is actually referenced in the introduction), directly visualizes this phenomenon. It explicitly shows vanilla CLIP attending to irrelevant background regions (e.g., mug instead of lamp, the ball instead of the flower), while DiffCLIP correctly focuses on the query-relevant objects. This is direct, visual proof of the attention noise problem we aim to solve.
>
> - **Quantitative Evidence:** Section 4.4 and Figure 4 present a rigorous quantitative evaluation on the MMVP benchmark, which is designed specifically to test fine-grained visual understanding. Our +5.7% absolute gain on this benchmark is strong quantitative evidence that reducing attention noise leads to improved performance on the very fine-grained tasks we mention in the introduction.
>
> We will ensure the text explicitly connects these results back to our initial motivation in the introduction to make the link undeniable for the reader.
>
> ---
>
> **Experimental evaluation is "rather superficial."**
>
> We believe this mischaracterizes the depth and breadth of our analysis. Our evaluation is intentionally multifaceted to provide a comprehensive picture of DiffCLIP's impact, going far beyond average accuracy.
>
> - **Breadth of Evaluation:** We test our method across six distinct task categories (linear probing, few-shot, retrieval, ZS, OOD, fine-grained) on over a dozen datasets. This demonstrates the generality and robustness of the improvement.
>
> - **Ablation Studies:** We provide two ablation studies that offer deeper insights: Section 4.6 (DiffCLIP$^\dagger$) isolates the source of the improvement, revealing that most gains come from the vision encoder alone. This is a non-obvious and highly valuable insight for the community. Section 4.5 (DiffCLIP*) explores the sensitivity to the `lambda_init` hyperparameter, uncovering a performance trade-off that contrasts with findings in the original NLP paper, highlighting modality-specific differences.
>
> Regarding the drop in performance on Aircraft and CIFAR - it is common not to have improvements on every single finegrained dataset (e.g SynthCLIP, LaCLIP, StableRep, ... are all papers were the same finegrained datasets were considered yet improvement of the proposed solution is not ALWAYS the best but is better overall).
>
> It is crucial to note that these are minor exceptions in a trend of overwhelmingly positive results across a vast array of benchmarks.
>
> ---
> 3- **Figure 2 and Table 3 are not referenced**
>
> While Table 3 was not referenced - it was directly presented with the results it entails in the same subsection. Figure 2 was referenced in the introduction. We will revise the manuscript to add a reference to Table 3 and speak more about Figure 2 in the introduction.
>
> ---
>
> 4-  **Analysis of the modality gap is missing.**
>
> While the modality gap is an important and fascinating area of research in VLMs, it is orthogonal to the central hypothesis of our paper. Our work investigates the impact of intra-modal attention noise within the encoders. The modality gap, as studied in the cited works, is an issue related to the inter-modal alignment of the final embeddings in the shared space.
>
> ---
>
> 5- **"Requested Changes" and "Minor" Points**
>
> - We will number the equations as requested by the reviewer.
> - We will provide full tables of the results instead of bar plots.
>
> ---
>
> **References:**
>
> [A] Greg Kamradt. Needle in a Haystack - pressure testing LLMs
>
> [B] Lost in the middle: How language models use long contexts
>
> [C] MaskCLIP: Masked Self-Distillation Advances Contrastive Language-Image Pretraining
>
> ---
>
> We hope this addresses the reviewers concerns fully.

---

> > ### Author Response · Authors · 2025-07-08
> >
> > We would like to invite the reviewer for further discussion to clarify any remaining concerns, exchange perspectives, and work towards enhancing the quality of the work.

---

> > > ### Comment · Reviewer_mLBQ · 2025-07-09
> > >
> > > Thank you for your response.
> > > I find your rebuttal mostly convincing and appreciate the clarification. However, I still have some concerns regarding how the attention noise is mitigated.
> > >
> > > While I agree that DiffCLIP improves the accuracy of the attention maps, the claim would be more strongly supported by including additional examples similar to Fig. 2, along with a quantitative comparison of attention accuracy.
> > > For instance, evaluating the attention maps using object detection or semantic segmentation datasets could help quantify how well the attention highlights the correct objects.

---

> > > > ### Author Response · Authors · 2025-07-10
> > > >
> > > > Thank you for your response and for finding our rebuttal mostly convincing, we appreciate this constructive discussion.
> > > >
> > > > ---
> > > >
> > > > 1- **Additional Qualitative Examples:**
> > > >
> > > > We have updated our manuscript's appendix to include four new, diverse examples that compare the attention maps of vanilla CLIP and our DiffCLIP. These additional visualizations consistently demonstrate the same pattern observed in Figure 2: DiffCLIP produces sparser, more object-centric attention maps that better align with the textual query, providing further qualitative support for our claim.
> > > >
> > > > ---
> > > >
> > > > 2- **On Quantifying the Attention Maps:**
> > > >
> > > > The main problem is that turning a fuzzy attention map into a clean, binary mask for an IoU score is really tough. You'd have to pick a threshold, but what's the right one? The distribution of the attention score for small objects is significantly different than big objects, so a single threshold wouldn't work for everything. We'd end up having to hand-tune it for every image, which feels a bit arbitrary and biased.
> > > >
> > > > Ultimately, what we care about is whether better attention actually helps the model do its job better. And our results show that it does. The +5.7% accuracy boost on the MMVP benchmark is probably the best quantitative proof we have. That benchmark is all about fine-grained questions that a model can only answer if it's really paying attention to the right details. So, seeing a big jump there tells us our approach is working as intended. We feel that measuring this real-world impact is more meaningful than trying to score the attention maps in isolation.
> > > >
> > > > Additionally, according to the best of our knowledge, we have not seen previous works report IoU or any sort of other metric to compare attention maps with segmentation maps.
> > > >
> > > > ---
> > > >
> > > > We hope this solves your concerns and convinces you to change `Claims And Evidence: No` to a Yes.

---

> > > > > ### Comment · Reviewer_mLBQ · 2025-07-15
> > > > >
> > > > > Thank you for your further response.
> > > > > I understand that proper thresholding for binarizing attention maps is difficult, and that improvement in MMVP implies more accurate attention maps.

---

> > > > > > ### Author Response · Authors · 2025-07-15
> > > > > >
> > > > > > Thank you very much for all the feedback.
> > > > > >
> > > > > > We are glad we could resolve your concerns. We will propagate your feedback into the manuscript !

---

### Review · Reviewer_gGUL · 2025-06-06

**Summary Of Contributions:**

This paper introduces DiffCLIP, a novel modification of the CLIP architecture that integrates a differential attention mechanism into both its vision and text encoders. The motivation stems from the observation that standard CLIP attention can focus on noisy or irrelevant features. Differential attention, adapted from recent work in large language models, aims to mitigate this by learning two complementary attention distributions and subtracting one from the other, thereby amplifying salient features and suppressing noise. The primary contribution is the successful adaptation and evaluation of differential attention for multimodal CLIP-style architectures, offering a lightweight method to enhance feature representation and task performance.

**Audience:**

Yes

**Claims And Evidence:**

Yes

**Requested Changes:**

While the paper cites Ye et al. (2024) and provides the core equations, a slightly more intuitive explanation of why subtracting one learned attention map from another helps cancel noise (e.g., what characteristics make the second map "complementary" or "noisy") could be beneficial for readers not deeply familiar with the original work.

The pretrained models and training code can be released, or add a statement, perhaps in the conclusion or as a footnote, indicating the authors' intent to release the code for DiffCLIP. They would be helpful for the community.

**Strengths And Weaknesses:**

Strengths:

The core idea of adapting differential attention to a dual-encoder VLM like CLIP is novel and well-motivated. The paper clearly articulates its goal: to reduce attention noise and improve multimodal representations.

A major strength is the minimal overhead. Achieving consistent performance gains with only a 0.003% increase in parameters and negligible computational cost makes DiffCLIP a very practical and appealing modification.

The authors evaluate DiffCLIP on a wide array of tasks: linear probing, few-shot classification, image-text retrieval (Flickr30k, Flickr8k, MSCOCO), zero-shot ImageNet, and out-of-distribution (OOD) robustness (ImageNet-V2, -A, -R, -Sketch). They also include fine-grained visual understanding using the MMVP-VLM benchmark. Experiments are conducted on two different pretraining dataset scales (CC3M and CC12M), demonstrating consistency. Across the board, DiffCLIP generally outperforms the standard CLIP baseline. While some individual gains are modest, the consistency across diverse tasks and datasets is a strong indicator of the method's effectiveness. The OOD and MMVP results are particularly noteworthy, suggesting better generalization and focus.

The paper is generally well-written, clearly explaining the motivation, the differential attention mechanism in the context of CLIP, the experimental setup, and the results. Figures and tables are informative.




Weaknesses:

The preliminary experiments integrating DiffCLIP's vision encoder into an LLaVA-style model (on the POPE benchmark) hint at the potential for this technique to benefit larger, more complex multimodal systems. More explorations can be done in the future.

The experiments use CC3M (2.3M pairs) and CC12M (7.9M pairs). While standard, this is smaller than the original CLIP's private 400M dataset. It's a limitation (common to many academic works) that the full potential at massive scale remains an open question, though the positive trend is encouraging. This is more of a scope limitation than a flaw.

While the paper cites Ye et al. (2024) and provides the core equations, a slightly more intuitive explanation of why subtracting one learned attention map from another helps cancel noise (e.g., what characteristics make the second map "complementary" or "noisy") could be beneficial for readers not deeply familiar with the original work.

---

> ### Author Response · Authors · 2025-07-03
>
> We are grateful to the reviewer for their detailed and positive assessment of our work. We are pleased they found our core idea to be "novel and well-motivated", the minimal overhead to be a "major strength", and our evaluation to be comprehensive and convincing, particularly noting the "consistency across diverse tasks".
>
> We will address the minor weaknesses and requested changes.
>
> ---
>
> 1- **On Expanding to Larger Models (LLaVA-style):**
>
> We thank the reviewer for recognizing the potential of our method. Our preliminary experiment in Section 5.1 was indeed intended to provide an initial signal that DiffCLIP-trained encoders could benefit larger vision-language models. We agree this is a compelling future direction. As the reviewer correctly intuits, a truly rigorous evaluation would require pre-training our vision encoder on a massive scale (e.g., LAION-400M), which is unfortunately beyond the computational capabilities of our academic lab. We will clarify this practical constraint in Section 5.1 of the final paper while highlighting this as a key avenue for future work.
>
> ---
>
> 2- **On the Scale of Pre-training Data:**
>
> The reviewer correctly notes that our experiments are conducted on CC3M and CC12M. We acknowledge this as a practical limitation inherent to academic research. However, we want to emphasize that our experimental scale is aligned with recent, impactful works from the academic community such as TripletCLIP (NeurIPS 2024), DreamLIP (ECCV 2024), and SynthCLIP, which relied on CC3M and CC12M as pretraining datasets for their CLIP experiments.
>
> ---
>
> 3- **On Providing a More Intuitive Explanation of Differential Attention:**
>
> This is an excellent suggestion. We agree that a more intuitive explanation would improve the paper's clarity.
>
> To summarize for the reviewer, the core idea is that the two attention maps are learned concurrently to be complementary. Let's revisit the core equation: $DiffAttn(X) = (A_1 - \lambda*A_2) * V$.
>
> - The first attention map, $A_1 = softmax(Q_1*K_1^T / \sqrt{(d/2)})$, is driven by the primary contrastive loss to focus on the most salient, task-relevant features (e.g., the "dog" in an image).
>
> - The second map, $A_2 = softmax(Q_2*K_2^T / \sqrt{(d/2)})$, is also trained via the same objective but with a separate set of projections. Through the subtraction, it is implicitly encouraged to capture more diffuse, high-entropy patterns that represent "noise", such as common background textures (e.g., "grass," "sky") or features that are spuriously correlated with the main object but not essential for identification.
>
> - By subtracting a scaled version of $A_2$ from $A_1$, the model effectively learns to cancel out this attentional noise. The final attention distribution becomes sparser and more focused on the truly discriminative features, leading to the improved performance and robustness we observe.
>
> We will incorporate this more detailed, intuitive explanation into Section 3.2.
>
> ---
>
> 4- **On Releasing Code and Models**
>
> We are fully committed to open science and reproducibility. All code and pretrained model checkpoints used in this work have been prepared for release and will be made publicly available upon publication.

---

> > ### Author Response · Authors · 2025-07-08
> >
> > Please let us know if our answer resolved the concerns you had regarding our work !

---

### Review · Reviewer_wZrT · 2025-07-03

**Summary Of Contributions:**

This paper proposes DiffCLIP, an extension of the Contrastive Language-Image Pre-training (CLIP) model where self-attention blocks are replaced by differential attention modules.
Differential attention is a previously published attention layer that was developed for LLMs and which produces attention maps consisting of the difference between two regular attention maps, a construction which was motivated by the observation that regular attention maps often contain noise that can be mitigated by the subtraction operation.
DiffCLIP is constracted in such a way that the additional number of parameters added to the corresponding CLIP model is negligible, and the computational overhead is also minimal.
Despite this minimal increase in parameters and computation cost, the paper demonstrates that the proposed method delivers consistent performance gains across classification, retrieval, robustness, and fine-grained vision benchmarks, although these gains are generally quite modest.

**Audience:**

Yes

**Broader Impact Concerns:**

nothing

**Claims And Evidence:**

No

**Requested Changes:**

* Provide uncertainty estimates (e.g. error bars) around the reported results to establish statistical significance of the performance gains, for instance by running multiple trials with different random seeds and reporting the standard deviation of the results, or by establishing some other null hypothesis testing procedure.

**Strengths And Weaknesses:**

### Strengths
* The motivation for the use of differential attention in CLIP models is straight-forward given the Differential Transformer paper
* Performance gains seem remarkably consistent across a variety of benchmarks, which is a strong point in favor of the proposed method

### Weaknesses
* The downside of such a straight-forward application of differential attention is that it makes for an arguably incremental contribution which might have limited interest to the community
* A major methodological issue of this paper is that it does not provide error bars around the reported results to establish statistical significance of the performance gains. This is a common problem in applied machine learning that fortunately the community is striving to overcome (the latest NeurIPS guidelines for example explicitly require reporting of error bars). This is particularly problematic specifically for this paper given that the performance gains are generally quite modest, making it very unclear whether they are statistically significant or not. And in general, as the community is moving towards more rigorous standards for reporting results, not reporting uncertainty estimates should start to be flagged as a major methodological flaw.

---

> ### Author Response · Authors · 2025-07-03
>
> We thank the reviewer for their thoughtful feedback and for recognizing the "remarkably consistent performance gains across a variety of benchmarks" as a strong point in favor of our method. We will address the two main weaknesses raised.
>
> ---
>
> **On the Contribution Being "Arguably Incremental"**
>
> We respectfully disagree with the characterization of our work as a "straight-forward application" with potentially "limited interest." While we build upon an existing mechanism, we believe our contribution is significant and multi-faceted, providing novel insights specifically for the vision-language domain.
>
> - **Demonstrating Efficacy in a New Domain:** Our work is the first to investigate and validate that a noise-cancellation mechanism from NLP can be successfully adapted to a dual-encoder vision-language architecture. The successful transfer of techniques across modalities is not a foregone conclusion; many NLP methods do not yield benefits in vision, and vice versa.
>
> - **Insights Through Ablations:** Our contribution is not just in the final model, but in the new understanding we explore through several ablations and evaluation benchmarks we tried out:
>   - **Where does the benefit come from?** Our ablation in Section 4.6 (DiffCLIP$^\dagger$) answers this by applying differential attention to the vision encoder alone. The finding that this configuration captures the vast majority of the performance gains is a critical, non-obvious insight. It strongly suggests that attention noise in the visual stream is a more significant bottleneck for CLIP-style models and provides a clear, cost-effective path for future VLM improvements.
>   - **How sensitive is the mechanism?** Our ablation in Section 4.5 (DiffCLIP*) investigates the impact of the initialization schedule for the key hyperparameter `lambda_init`. We compare a static vs. a dynamic, layer-dependent schedule, revealing a trade-off where the dynamic schedule boosts certain zero-shot tasks at the expense of others. This is contrary to the observation by the authors in Differential Transformer where dynamic was always better.
>   - **What is the qualitative effect?** Our analysis in Figure 2 and the quantitative results on the fine-grained MMVP benchmark (Figure 4, a +5.7% absolute gain) provide concrete evidence that the mechanism directly improves the model's ability to focus on salient objects and suppress background noise.
> ---
>
> 2-**On the Lack of Error Bars and Statistical Significance**
>
> We agree with the reviewer that reporting uncertainty estimates is a vital component of rigorous empirical research.
>
> The reviewer's concern is entirely valid in principle. However, it must be weighed against the practical realities and computational costs of large-scale pre-training. As an academic institution, pre-training a single CLIP-B/16 model on CC3M (2.3M pairs for 40 epochs on 4 A100s) requires approximately 90 A100 GPU-hours. Running this 3-5 times for each of our model variants (CLIP, DiffCLIP, DiffCLIP*, DiffCLIP†) to generate standard deviations is computationally prohibitive and would require thousands of GPU-days, a resource scale that is unfortunately beyond our reach.
>
> Despite this constraint, we argue that the statistical significance of our results is strongly supported by an alternative form of evidence: *the consistency and breadth of the improvements.*
>
> - Our method is not tuned for a single task. We report results across six distinct evaluation categories (linear probing, few-shot, image retrieval, text retrieval, ZS ImageNet, ZS OOD) and over a dozen individual datasets.
>
> - DiffCLIP shows an improvement over regular CLIP in the vast majority of the evaluations. The probability of achieving such a consistent pattern of positive results across such a diverse suite of tasks by random chance is very low. This pattern provides good evidence that the underlying effect is real and not an artifact of a particular random seed.
>
> **Note to Reviewer:** Alternatively, we could run the evaluations such as few-shot and linear probing with different seeds as they are non-deterministic evaluations unlike retrieval tasks and MMVP.
>
> ---
>
>
> We hope this answers the concerns of the reviewer.

---

> > ### Comment · Reviewer_wZrT · 2025-07-03
> >
> > Dear Authors,
> >
> > Thank you very much for you answers. I apologize if "straight-forward" sounded belittling. As a straight-forward, direct and clear application of a previous architectural innovation to a novel setting and class of models I personally find your work valuable and insightful. I was, maybe with an excess of caution, trying to gauge the broader appeal to the community.
> >
> > As for providing error bars, I understand that repeating all experiments on multiple seeds could be computationally taxing. But there are other ways as well to assess statistical significance of results. First of all, one possibility that is sometimes a good alternative (you'll have to determine for yourself if this is a meaningful suggestion in your case) is to assess statistical significance with smaller models and datasets, to at least demonstrate that the stated claims of superiority in performance are at least rigorously supported in that setting. Alternatively, rejecting a weaker statistical hypothesis than (in your case) showing that DiffCLIP seeds lead to better trained models than CLIP seeds might be an appropriate strategy to consider. For instance, one could perform a permutation test to determine whether the test set accuracy of the one DiffCLIP checkpoint that you trained is significantly better than that of the CLIP checkpoint that you trained. It's a much weaker test, but at least it would give actually allow you to state a rigorously supported claim.
> >
> > It's been a long time coming for the community to uphold standards of rigor and reproducibility, as it has recently begun to do, for instance, with the NeurIPS guidelines, particularly the request to provide statistical significance for experiments. I strongly oppose renegotiating and reversing this type of progress.

---

> > > ### Author Response · Authors · 2025-07-03
> > >
> > > Thank you for your thoughtful follow-up and for clarifying your perspective. We appreciate you acknowledging the value and insight of our work, and we want to engage seriously with your important point about statistical rigor.
> > >
> > > We completely agree with the principle you are advocating. The community's push towards higher standards of rigor and reproducibility is a positive and necessary development. To that we are committed to releasing our code and models. Our disagreement is not with the principle, but with its application in the specific context of large-scale vision-language pre-training, where the computational cost is a fundamental constraint for academic labs.
> > >
> > > **On alternative statistical tests.** While we appreciate these constructive suggestions, we must argue that the most rigorous and accepted form of validation in this specific research area is precisely the one we have employed. The established standard for large-scale VLM pre-training in top-tier conferences is to demonstrate overwhelming consistency across a wide array of diverse tasks when multi-seed runs are infeasible. This is not a standard we are proposing; it is the one currently in practice. As direct evidence, we point to several highly-regarded, contemporary papers that follow the same evaluation strategy as we do and not perform multiple seeds or statistical testing:
> > >
> > > - TripletCLIP (NeurIPS 2024)
> > > - DreamLIP (ECCV 2024)
> > > - SynthCLIP (CVPRW 2024)
> > > - StableRep (NeurIPS 2023)
> > > - FineLIP (CVPR 2025)
> > > - FLAME (CVPR 2025)
> > > - Tulip (2025)
> > > - Advancing Compositional Awareness in CLIP with Efficient Fine-Tuning (2025)
> > >
> > > None of these papers, accepted at the field's most prestigious venues, report multi-seed pre-training runs or the post-hoc statistical tests you suggest. **_Instead, they rely on the same evidence we do: demonstrating that their proposed method consistently outperforms strong baselines across a comprehensive suite of benchmarks._**
> > >
> > > The suggestion to run permutation tests on our single checkpoints is, as the reviewer notes, a much weaker statistical claim. Such a test would only assess whether the performance difference on a specific test set is likely due to the random partitioning of that set, not whether our training method is fundamentally superior. The central question of our paper is whether DiffCLIP as a training procedure produces better models than standard CLIP. This question can only be truly answered by comparing multiple training runs.
> > >
> > > If we had access to large computational resources, we would, without hesitation, run multiple full pre-training experiments from scratch. This is the gold standard. However, since this is infeasible, the next best form of evidence is not a weaker statistical test on a single run, but rather the robustness of the performance gains across a vast and diverse set of downstream tasks. To demand a different form of testing would be to hold our paper to a standard that is not currently applied to any other work in this specific subfield.
> > >
> > > ---
> > >
> > > To receive `Claims And Evidence: No` when the only concern stems from a request for a procedure that is not standard practice in this area of research feels a bit unfair.

---

> > > > ### Author Response · Authors · 2025-07-08
> > > >
> > > > We would like to invite the reviewer for further discussion to clarify any remaining concerns, exchange perspectives, and work towards enhancing the quality of the work.

---

> > > > > ### Comment · Reviewer_wZrT · 2025-07-15
> > > > >
> > > > > The question about the statistical significance of the reported results is still outstanding. None of the quantitative claims of the paper are supported by results that are backed by rigorous statistical analysis. Without clear evidence of statistical significance, it is challenging to draw reliable conclusions from the findings, particularly in the current situation where the reported performance results are so close to the baseline. Despite the authors attempts at trying to argue their way out the effort of complementing their results with statistical rigor, the importance of that in terms of establishing the validity of results, reproducibility, transparency and basic understanding is something that is fortunately starting to permeate the community. The few papers cited by the authors as counterexamples are not indicative of where the community is attempting to go, trivializes and contributes to hindering this collective effort, and also minimizes the contribution of the mentioned papers themselves.

---

### Decision · Action_Editor_2Mdj · 2025-08-02

**Recommendation:** Accept with minor revision

**Additional Comments:**

The article got mixed scores. Reviewers appreciated the proposed technical change and its motivation (mLBQ, gGUL, wZrT), finding the experiments extensive (mLBQ, gGUL, wZrT). While the response and revision addressed mLBQ and gGUL's major concerns (clarifications, scale, analyses), wZrT raised concerns on the significance of the results, given the limited gaps with the base CLIP and the lack of error bars. Moreover, mLBQ points to the lack of quantitative evaluation of the attention maps as something missing to more thoroughly substantiate the paper's claims.

The AE went over the paper, rebuttal, and responses finding the article valuable for the community and deserving acceptance. At the same time, the AE is aligned wZrT on the utmost importance of providing error bars to add more evidence to the claims. At the same time, I understand the authors' perspective and the difficulty on providing multiple pretraining runs given their large computational cost. As a middle ground, the final version of the work should provide results on multiple seed, together with error bars, on the non-deterministic experiments (linear probing and few-shot), as proposed also by the authors in their first reply. Beyond this, the revision should discuss limitations/difficulties in evaluating the attention maps (as per the point raised by mLBQ).

**Audience:**

Yes

**Audience Explanation:**

As this paper proposes a modification of the widely used CLIP, it is of interest to all researchers working on building better vision-language representations and models.

**Claims And Evidence:**

Yes

**Claims Explanation:**

The article proposes a variant of CLIP based on Differential Attention, replacing self-attention blocks with differential attention ones. The main intuition behind this choice, is that the model would be able to filter out noise and amplify relevant content. The efficacy of the approach is evaluated against base CLIP on a wide range of setting showing consistent improvements, thus validating its efficacy (despite the relatively close margins).

---

> ### Author Response · Authors · 2025-08-13
> **Thank You !**
>
> Thank you very much for accepting our paper and for your constructive feedback throughout the process. We also sincerely thank all the reviewers for their valuable comments and suggestions. We have now submitted the camera-ready version with all requested changes. Please let us know if there is anything else needed from our side.